# Physical restraint of dementia patients in acute care hospitals during the COVID-19 pandemic: A cohort analysis in Japan

**Takuya Okuno**, **Hisashi Itoshima**, **Jung-ho Shin**, **Tetsuji Morishita**, **Susumu Kunisawa**, **Yuichi Imanaka** *

Department of Healthcare Economics and Quality Management, Graduate School of Medicine, Kyoto University, Kyoto City, Kyoto, Japan

* imanaka-y@umin.net

## Abstract

### Introduction

The coronavirus disease (COVID-19) pandemic has caused unprecedented challenges for the medical staff worldwide, especially for those in hospitals where COVID-19-positive patients are hospitalized. The announcement of COVID-19 hospital restrictions by the Japanese government has led to several limitations in hospital care, including an increased use of physical restraints, which could affect the care of elderly dementia patients. However, few studies have empirically validated the impact of physical restraint use during the COVID-19 pandemic. We aimed to evaluate the impact of regulatory changes, consequent to the pandemic, on physical restraint use among elderly dementia patients in acute care hospitals.

### Methods

In this retrospective study, we extracted the data of elderly patients (aged > 64 years) who received dementia care in acute care hospitals between January 6, 2019, and July 4, 2020. We divided patients into two groups depending on whether they were admitted to hospitals that received COVID-19-positive patients. We calculated descriptive statistics to compare the trend in 2-week intervals and conducted an interrupted time-series analysis to validate the changes in the use of physical restraint.

### Results

In hospitals that received COVID-19-positive patients, the number of patients who were physically restrained per 1,000 hospital admissions increased after the government's announcement, with a maximum incidence of 501.4 per 1,000 hospital admissions between the 73rd and 74th week after the announcement. Additionally, a significant increase in the use of physical restraints for elderly dementia patients was noted ($p = 0.004$) in hospitals that received COVID-19-positive patients. Elderly dementia patients who required personal care experienced a significant increase in the use of physical restraints during the COVID-19 pandemic.

**Data Availability Statement:** According to the Ethical Guidelines for Medical and Health Research Involving Human Subjects of the Ministry of Health, Labour and Welfare, Japan (a provisional

translation is available from https://www.mhlw.go.jp/file/06-Seisakujouhou-10600000-Daijinkanboukouseikagakuka/0000080278.pdf) , providing "information" to study subjects is one of the necessary conditions for waiving informed consent. The range of data users is included the "information". Therefore, the datasets generated during and/or analyzed during this study are available from the corresponding author and Office of Research Promotion, General Affairs and Planning Division, Kyoto University (E-mail: kikaku06@mail2.adm.kyoto-u.ac.jp; Tel: +81-75-753-9301) on reasonable request. The Ethics Committee, Graduate School of Medicine, Kyoto University (e-mail: ethcom@kuhp.kyoto-u.ac.jp) also can help those who need further information regarding the data availability of this study.

**Funding:** This study was supported by JSPS KAKENHI Grant Number JP19H01075 from the Japan Society for the Promotion of Science, GAP Fund Program of Kyoto University type B, Health Labour Sciences Research Grant from the Ministry of Health, Labour and Welfare, Japan [21IA1005], and Humanities, Social/Behavioral Sciences, and Natural Sciences Interdisciplinary Research Project of Kokoro Research Center from Kyoto University to Y.I. The funders played no role in the study design, data collection and analysis, decision to publish, or preparation of the manuscript.

**Competing interests:** The authors have declared that no competing interests exist.

## Conclusion

Understanding the causes and mechanisms underlying an increased use of physical restraints for dementia patients can help design more effective care protocols for similar future situations.

## Introduction

The rapid spread of coronavirus disease (COVID-19) has progressively increased and continues to disrupt healthcare systems worldwide [1]. In acute care hospitals, especially those treating COVID-19 patients, the medical staff members face difficulties in providing routine care owing to patient triage, social distancing, and shortage of resources such as finances, medical supplies, and manpower [2–4]. To manage the pandemic, the Japanese government announced hospital restrictions, including those pertaining to family visits, at the end of March 2020. Eventually, a state of emergency was declared for specific areas on 7[th] April, 2020, and implemented nationwide on 16[th] April, 2020. Social distancing and limiting family visits impacted the hospital care systems in many ways, such as reduced communications with medical staff and family members; these in turn could exacerbate progressive cognitive dysfunction and worsen behavioral and psychological symptoms in dementia patients and consequently, result in higher distress to both patients and medical staff [5].

The use of physical restraint for dementia patients has been discussed in recent years. Physical restraint is often used in acute care settings [6–8] and includes 11 means of mechanical restraint based on national guidelines for the prevention of physical restraints [9]. However, such means may confer critical medical disadvantages for patients, including restraint device-related injuries, such as asphyxiation or chest compression, and immobility-related complications, such as deep vein thrombosis, pulmonary embolism, aspiration pneumonia, and rhabdomyolysis [10–13].

Owing to the abovementioned disadvantages and ethical concerns, recommendations to avoid the use of physical restraint have been made worldwide, including Japan [14–16]. Since 2016, in Japan, the Ministry of Health, Labour and Welfare (MHLW) has factored in an additional fee in the universal benefit scheme for dementia care of patients without severe disorientation who need personal care, wherein a financial disincentive of 40% reduction is provided if physical restraint is used [17]. As dementia symptoms may not be recognizable during routine care, this benefit may be particularly applicable to patients who have communication-related challenges or symptoms that inhibit their daily life without diagnosis of dementia [18]. To be eligible to obtain the stipulated benefit, nurses need to be trained in dementia care, and a standardized protocol for mechanical and chemical restraint procedures for sedation is required [17].

Providing routine comprehensive care for dementia patients may have been especially challenging during the 1st wave of the COVID-19 pandemic as the unprecedented crisis seriously impacted the healthcare systems [5]. However, only few studies have explored the impact of the COVID-19 pandemic on dementia care, especially with regard to physical restraint use for dementia patients. Therefore, in this retrospective cohort study, we aimed to evaluate the changes in the use of physical restraints among dementia patients in acute care hospitals stratified on the basis of them receiving or not receiving COVID-19-positive patients. We hypothesized that dementia patients are more likely to be physical restrained during the pandemic in acute care hospitals that treat COVID-19 patients than during pandemic-free time periods.

## Materials and methods

### Data source

We used the Diagnosis Procedure Combination (DPC) data from the Quality Indicator/
Improvement Project (QIP) database in Japan. The QIP participant hospitals provide claims
data and DPC data to improve their system and quality of care using quality indicators. Across
Japan, more than 200 QIP participant hospitals, both public and private, and of various sizes
were included; in these hospitals, the number of general beds (hospital beds not earmarked as
psychiatric, infectious disease, and tuberculosis beds) according to the Japanese classification
of hospital beds ranged from 30 to 1,151 in 2019.

The DPC/per-diem payment system (PDPS) is a Japanese prospective payment system that
is used in acute care hospitals and is comparable to diagnosis-related databases in the United
States [19, 20]. A total of 1,730 hospitals adopted the DPC/PDPS in 2018, which accounted for
54% of all general beds in Japanese hospitals [21, 22]. However, the DPC data do not include
detailed information on the level of nursing care; instead, they provide information such as
primary diagnoses, comorbidities (identified using the International Classification of Diseases,
10th Revision [ICD-10] codes), drug or device prescriptions, and codes corresponding to the
performed medical procedures as stated in the discharge summary.

### Study population

The eligibility criteria for inclusion in this study were as follows: age > 64 years; availability of
admission and discharge summary for 78 weeks between January 6, 2019, and July 4, 2020;
and application of dementia care benefit during admission. We excluded patients who were
admitted to the intensive care unit or were hospitalized for COVID-19 treatment because their
clinical characteristics and disease severity greatly differed from those of other dementia
patients, and therefore, the use of physical restraints in the former could be a consequence of
other factors/mechanisms.

### Variables

We obtained information on patients' sex, age, ambulance use, admission type, admission
pathway, comorbidity indices (Charlson Comorbidity Index [CCI]) [23], whether a surgical
procedure was conducted, reason for admission based on the ICD-10 codes (infection, neo-
plasm, endocrine, mental and behavioral, nervous, circulatory, respiratory, digestive, musculo-
skeletal, genitourinary, injury, and others), and length of stay (LOS) to examine the baseline
patient characteristics. The patients were assigned to three groups based on age (65–74, 75–84,
and $\geq$ 85 years). LOS is presented as the median and interquartile range. The outcome of
interest was the frequency of physical restraint use among patients who applied for dementia
care benefit. Data regarding the use of physical restraint during dementia care were extracted
from the payment codes for services.

### Statistical analysis

First, we divided the 78-week period into 39 categories of 2-week intervals based on the admis-
sion data and specified the appropriate timing category of the announcement of COVID-19
hospital restrictions by the Japanese government (33rd out of 39 categories) as the point of
implementation, the state of emergency. In Japan, hospitals that can accept COVID-19-posi-
tive patients were designated by the MHLW. If no COVID-19-positive patients were hospital-
ized during this study period, we considered that the impact of COVID-19 was small.
Therefore, we categorized the study population into two groups: hospitals having at least one

COVID-19-positive patient admission during the study period (Group 1) and those having none (Group 2). Subsequently, we divided our datasets into two periods for the interrupted time-series (ITS) analysis: pre- (1–32) and post-announcement (33–39). Comparisons were conducted using the chi-squared test or Fisher's exact test or the Kruskal–Wallis rank sum test, as appropriate. Thereafter, to examine the trend, the number of patients who were physically restrained per 1,000 hospitalizations as indicated for every 2 weeks during the whole study period are presented in line graphs for each group. Finally, we used ITS, including segmented regressions, to ascertain the impact of the government's announcement of the state of emergency. We statistically assessed the changes in the number of patients who were physically restrained per 1,000 hospitalizations and who were provided with the dementia care benefit based on the date of admission adjusted for seasonality through a Fourier term [24]. The level of statistical significance was set at $p < 0.05$ (two-tailed). Statistical analyses were performed with R version 4.0.2 (R Foundation for Statistical Computing, Vienna, Austria).

The study protocol was approved by the Ethics Committee, Kyoto University Graduate School and Faculty of Medicine. This study was conducted in accordance with the ethical guidelines issued by the Japanese National Government for medical and health research involving human participants. The data were anonymized, and the requirement for informed consent was waived by the approving authority.

## Results

We identified 158,797 admissions from 245 hospitals. After excluding patients admitted to the ICU (n = 2,737) and those being treated for COVID-19 (n = 204), 155,856 admissions were finally recognized. Thereafter, we divided the hospitals into 2 groups, admissions in hospitals that received COVID-19 positive patients (Group 1, 97,233 admissions) and those that received none (Group 2, 58,623 admissions), which are shown in Fig 1. Tables 1 and 2 show the demographics of patients who were eligible for inclusion from before to after the state of emergency announced by the MHLW in both groups. All variables are expressed as absolute numbers (n) and relative frequencies (%). Among the patients, those aged > 85 years comprised more than half of the study population, and most patients needed urgent or emergent hospitalization (Group 1: 86.6% vs. 86.9%; Group 2: 78.9% vs. 77.2%). In Group 1, the percentages of subjection to surgical procedures pre- and post-COVID-19-related regulatory intervention (11.7% vs. 12.1%; $p = 0.094$) were marginally high, whereas the percentage of CCI > 2 was low (19.8% vs. 18.5%; $p < 0.001$) after the intervention. Fig 2 displays the number of patients who were physically restrained per 1,000 hospitalizations (shown on the line) in both groups. Group 2 shows a lower number throughout the study period than Group 1. After the 66th week of announcement by the MHLW, the number of cases that required physical restriction per 1,000 hospitalizations increased, with a maximum of 501.4 during the 73rd and 74th week in Group 1. According to the ITS analysis in Fig 3, the number of patients who were physically restrained per 1,000 hospitalizations significantly increased only in Group 1 after the state of emergency was announced by the MHLW (Group 1: $p = 0.004$; Group 2: $p = 0.437$).

## Discussion

This study examined the trends of dementia patients requiring nursing care who were physically restrained per 1,000 hospitalizations and tracked important changes in this regard during the COVID-19 pandemic in Japan. The main finding of our study was that following the MHLW's announcement of COVID-19 hospital restrictions and the state of emergency, dementia patients who required nursing care were significantly more likely to be physically restrained in hospitals that received COVID-19-positive patients.

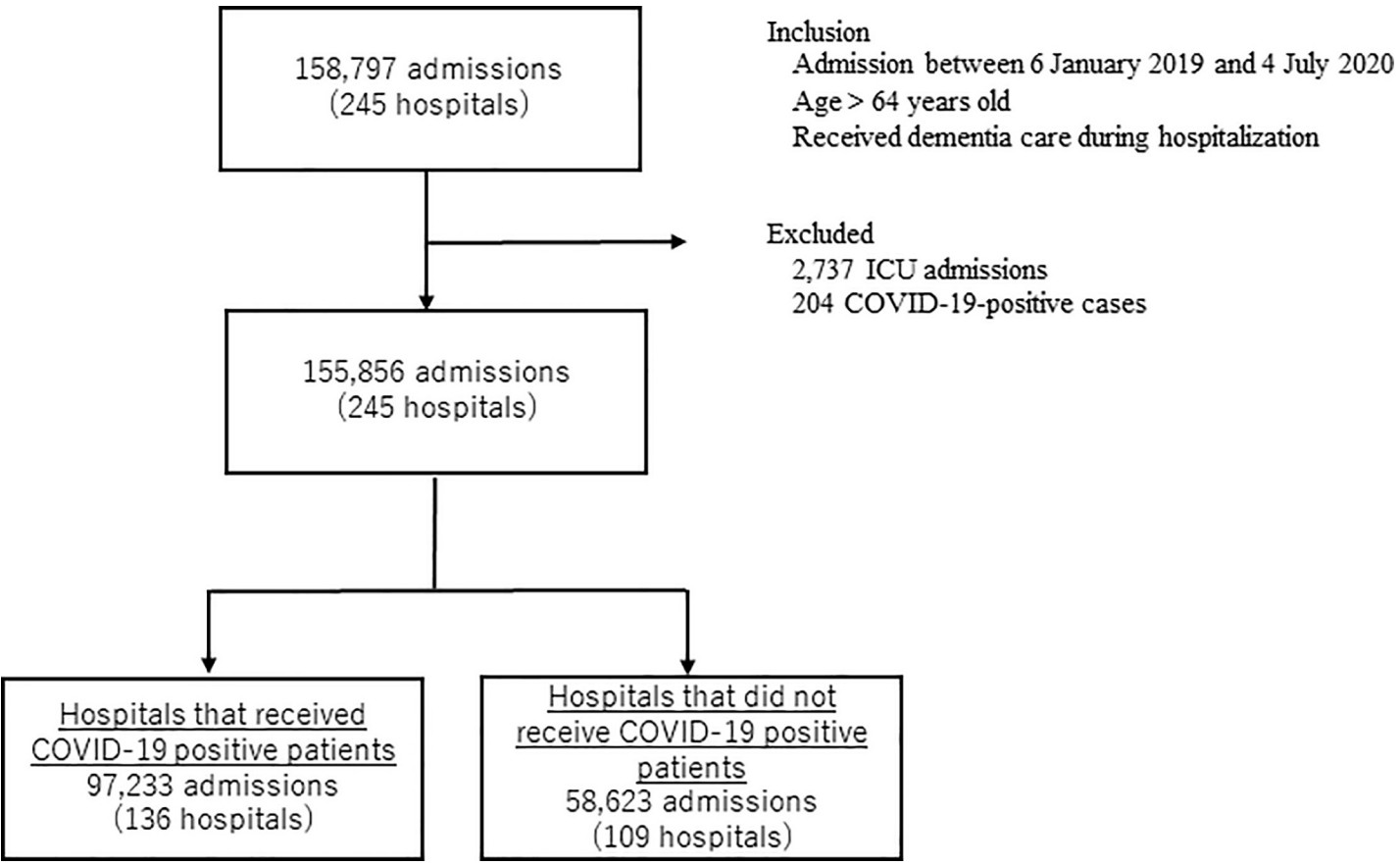

**Fig 1. Flowchart depicting patient progression in this study based on the eligibility and exclusion criteria.**

Dementia has increasingly gained importance as a public health concern, and the medical staff in acute care hospitals often need to provide dementia care to elderly patients [25, 26]. Physical restraint, which is preferably avoided wherever possible, in conformance with worldwide recommendations, is often exercised in acute care settings, especially for elderly patients and those with dementia [7, 8, 15, 27–29]. Physical restraint is exercised to prevent falls and self-extubation owing to the low availability of medical staff and inadequate resources to constantly monitor at-risk patients because of the immense workload [6, 30]. There are few reports about changes in the implementation rate of physical restraint due to disasters such as the COVID-19 pandemic. However, one recent observational study showed the possibility of increased use of physical restraint during the COVID-19 pandemic [31], which supports our results.

We believe that the main reason for the significantly increased use of physical restraints for elderly dementia patients in only Group 1 during the COVID-19 pandemic was due to factors associated with the quality of care. Although it has been reported that cognitive function of the elderly may worsen with social distancing being implemented nationwide in Japan [32, 33], the reason for an obvious increase in the use of physical restraint in the ITS analysis at the hospitals without any hospitalization of COVID-19-positive patients in this study was unclear.

The mental and physical statuses of the medical staff are important to provide the best care for patients. During the COVID-19 outbreak, the medical staff were under pressure owing to the heavy workload and higher risk of infection due to lack of sufficient personal protective

**Table 1. Characteristics of patients who received dementia care before and after COVID-19-related regulatory changes in hospitals that received COVID-19-positive patients.**

| Characteristics | Pre-intervention | Post-intervention | p |
|---|---|---|---|
| Number of patients | 80,468 | 16,765 | |
| Male, n (%) | 34,410 (42.8) | 7,084 (42.3) | 0.23 |
| Age, years, mean (SD) | 85.01 (7.35) | 84.97 (7.31) | 0.611 |
| Age category, years, n (%) | | | 0.282 |
| 65–74 | 7,486 (9.3) | 1,538 (9.2) | |
| 75–84 | 27,775 (34.5) | 5,894 (35.2) | |
| ≥85 | 45,207 (56.2) | 9,333 (55.7) | |
| Ambulance use, n (%) | 40,509 (50.3) | 8,423 (50.2) | 0.823 |
| Urgent or emergent admission, n (%) | 69,646 (86.6) | 14,566 (86.9) | 0.252 |
| Admission pathway, n (%) | | | 0.545 |
| Home | 51,478 (64.0) | 10,651 (63.5) | |
| Hospital or nursing home | 28,932 (36.0) | 6,101 (36.4) | |
| Other | 58 (0.1) | 13 (0.1) | |
| Charlson Comorbidity Index >2, n (%) | 15,951 (19.8) | 3,104 (18.5) | <0.001 |
| Surgery during admission, n (%) | 9,377 (11.7) | 2,031 (12.1) | 0.094 |
| Reason for admission, n (%) | | | <0.001 |
| Infection, n (%) | 2,194 (2.7) | 457 (2.7) | |
| Neoplasm, n (%) | 5,336 (6.6) | 1,214 (7.2) | |
| Endocrine, n (%) | 4,047 (5.0) | 827 (4.9) | |
| Mental and behavioral, n (%) | 360 (0.4) | 68 (0.4) | |
| Nervous, n (%) | 2,579 (3.2) | 501 (3.0) | |
| Circulatory, n (%) | 15,087 (18.7) | 2,895 (17.3) | |
| Respiratory, n (%) | 16,829 (20.9) | 2,936 (17.5) | |
| Digestive, n (%) | 8,414 (10.5) | 1,947 (11.6) | |
| Musculoskeletal, n (%) | 2,187 (2.7) | 463 (2.8) | |
| Genitourinary, n (%) | 6,750 (8.4) | 1,587 (9.5) | |
| Injury, n (%) | 11,061 (13.7) | 2,490 (14.9) | |
| Others, n (%) | 5,624 (6.9) | 1,380 (8.0) | |
| Length of stay, median (IQR) | 21 [12, 39] | 20 [12, 36] | <0.001 |

SD: standard deviation, IQR: interquartile range

equipment [34–36]. Owing to the increase in nosocomial infections from February to April, 2020, medical staff were seen as epicenters, and this led to widespread irrational prejudice and discrimination against them in off duty-hours. They were denied use of public vehicles and their children were asked to refrain from attending nursery schools [36]. In hospital, nurses are required to take care of several patients simultaneously during pandemics, such as the COVID-19 pandemic [37], while wearing personal protective equipment, which makes communication difficult. The threshold for physically restraining elderly dementia patients may have been lowered owing to changes in the care system that have occurred consequent to the implementation of hospital strategies or owing to an increase in both physical and mental strain on medical staff.

Factors associated with the care system, including limiting family visits, might have also possibly affected the result. In Japan, even the state of emergency is not legally binding; therefore, the hospital visit restrictions at hospitals without COVID-19 positive patients might have been more permissive than hospitals with COVID-19-positive patients' hospitalizations. For

**Table 2. Characteristics of patients who received dementia care before and after COVID-19-related regulatory changes in hospitals that received no COVID-19-positive patients.**

| Characteristics | Pre-intervention | Post-intervention | P |
|---|---|---|---|
| Number of patients | 48,424 | 10,199 | |
| Male, n (%) | 19,746 (40.8) | 4,223 (41.4) | 0.245 |
| Age, years, mean (SD) | 85.51 (7.37) | 85.34 (7.35) | 0.029 |
| Age category, years, n (%) | | | 0.018 |
| 65–74 | 4,200 (8.7) | 919 (9.0) | |
| 75–84 | 15,305 (31.6) | 3,344 (32.8) | |
| ≥85 | 28,919 (59.7) | 5,936 (58.2) | |
| Ambulance use, n (%) | 17,349 (35.8) | 3,587 (35.2) | 0.214 |
| Urgent or emergent admission, n (%) | 38,216 (78.9) | 7,877 (77.2) | <0.001 |
| Admission pathway, n (%) | | | 0.826 |
| Home | 27,615 (57.0) | 5,850 (57.4) | |
| Hospital or nursing home | 20,789 (42.9) | 4,345 (42.6) | |
| Other | 20 (0.0) | 4 (0.0) | |
| Charlson Comorbidity Index >2, n (%) | 9,029 (18.6) | 1,907 (18.7) | 0.913 |
| Surgery during admission, n (%) | 3,715 (7.7) | 731 (7.2) | 0.084 |
| Reason for admission, n (%) | | | <0.001 |
| Infection, n (%) | 954 (2.0) | 177 (1.7) | |
| Neoplasm, n (%) | 2,257 (4.7) | 520 (5.1) | |
| Endocrine, n (%) | 2,370 (4.9) | 563 (5.5) | |
| Mental and behavioral, n (%) | 421 (0.9) | 102 (1.0) | |
| Nervous, n (%) | 1,936 (4.0) | 362 (3.5) | |
| Circulatory, n (%) | 9,504 (19.6) | 2,176 (21.3) | |
| Respiratory, n (%) | 10,760 (22.2) | 1,761 (17.3) | |
| Digestive, n (%) | 4,056 (8.4) | 900 (8.8) | |
| Musculoskeletal, n (%) | 1,827 (3.8) | 446 (4.4) | |
| Genitourinary, n (%) | 4,122 (8.5) | 910 (8.9) | |
| Injury, n (%) | 7,153 (14.8) | 1,604 (15.7) | |
| Others, n (%) | 3,064 (6.3) | 648 (6.4) | |
| Length of stay, median (IQR) | 25 [14, 50] | 26 [14, 50] | 0.579 |

dementia patients, communicating with visitors, especially family members, is important to maintain their cognitive function [29, 31, 38, 39]. The Centers for Disease Control and Prevention guidelines allow care partners to visit patients if they are essential to the patients' physical or emotional well-being, even during the COVID-19 pandemic [40]. Furthermore, use of telemedicine and digital technology can be helpful for the management of chronic neurological diseases, including dementia and cognitive impairment [41].

This study had several limitations. First, the severity of manpower shortage and the extent to which the restriction regarding family visitation was strictly enforced were unclear. More thorough infection control measures were considered to be practiced in hospitals that treated COVID-19-positive patients than in hospitals that did not. However, we could not consider and evaluate different burdens on the medical staff owing to differences in the number of admissions of COVID-19-positive patients in the target hospitals. To manage restrictions on in-person visits owing to the COVID-19 pandemic, some hospitals have been attempting to ensure a virtual connection between patients and their loved ones via tablets or smartphones. Despite the limitations in the use of technology, including difficulty in hearing over devices, patients can benefit by communicating with their family members [42]. Second, we could not

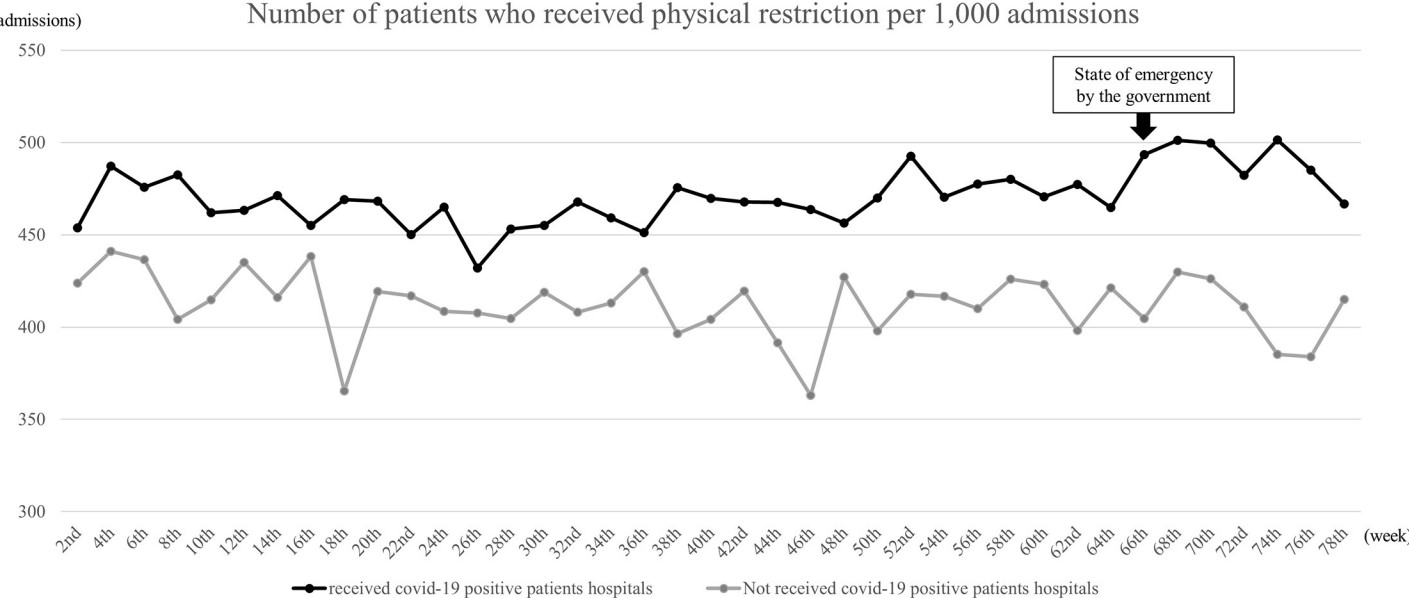

**Fig 2. Comparison of the number of patients restrained between the two groups.** The number of patients physically restrained per 1,000 hospital admissions for 2-week intervals between January 1, 2019, and June 30, 2020, in the two groups.

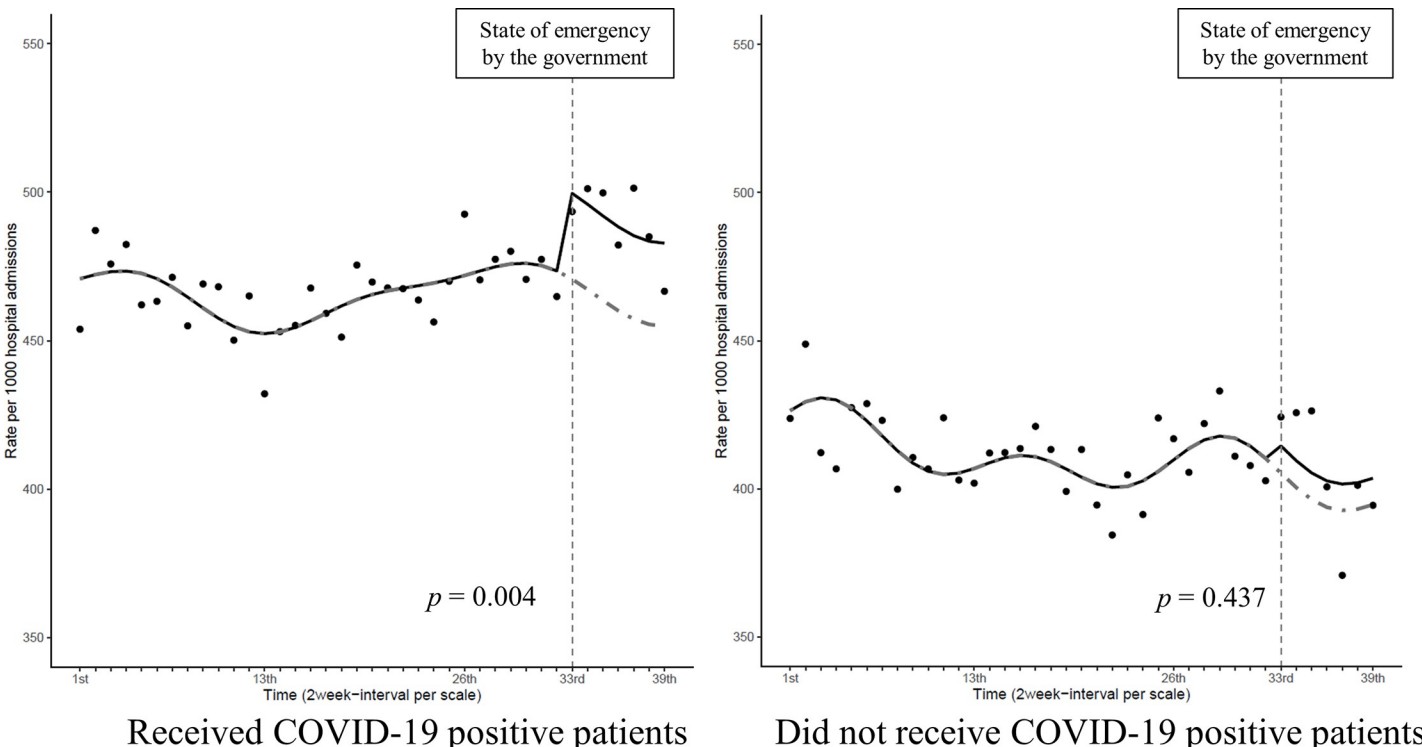

**Fig 3. Interrupted time-series analysis of the number of patients who were restrained.** The number of patients physically restrained per 1,000 hospital admissions over time was evaluated with an interrupted time-series analysis including segmented regressions (Group 1: $p = 0.032$; Group 2: $p = 0.341$). The solid line represents the actual transition and the dotted line represents the hypothetical transition in the absence of intervention.

evaluate the exact quality of dementia care in each group. As shown in Fig 2, hospitals with COVID-19-positive cases already have higher percentages of restraint in the hospital than those before the COVID-19 pandemic. It may be desirable to evaluate the quality of dementia care and the involvement of a geriatric specialist; however, this is not possible owing to a limited database. However, the target population for this study was the inpatients for whom dementia care benefit was calculated. We believe that the quality of care in the two populations is secured to a certain extent because the hospitals need to have staff trained in dementia care and conduct regular care meetings in order to calculate the additional fee. Additionally, since this study uses the impact of COVID-19 as an intervention point to compare the percentage of physical restraint practices in the hospital for each group over time, we believe that this is not a problem. Third, we could not detect the type or severity of dementia, which is often not recognized in general hospitals [18], as the applicable benefit did not require precise information about dementia. However, patients who were eligible for inclusion in this study were patients who were judged by the medical staff, trained in dementia care, as having dementia or an equivalent cognitive impairment that interfered with their daily lives and necessitated nursing care [17]. Moreover, the dementia care benefit cannot be applied to those who have severe disorientation (indicated with a Glasgow Coma Scale score < 9) [9, 17]. Therefore, we believe that patients with dementia of severity within a certain range were selected. Fourth, the proportions of each reason for hospitalization may differ before and after the COVID-19 pandemic. In this study, we attempted to evaluate the use of physical restraint to whole inpatient populations with cognitive impairment. Therefore, we could not fully consider the difference in the implementation rate of physical restraint use per disease.

However, we included large sample sizes, which is a strength of our study. Multicomponent interventions that increase medical staff awareness have limited effectiveness in reducing physical restraint use [16]; however, we believe that examining the current situation during the pandemic can significantly help prepare for similar future circumstances.

## Conclusions

We demonstrated and validated a trend of increased use of physical restraints for elderly dementia patients using ITS analyses of administrative data. Elderly dementia patients who require personal care might be more likely to be physically restrained during the COVID-19 pandemic in hospitals receiving COVID-19-positive patients. While limited social interaction is inevitable to prevent the spread of COVID-19, the promotion of telemedicine and mental or physical care for medical staff may be important in reducing the use of physical restraints among dementia care patients. Future research should identify causative factors, including patient environment and stress among medical staff members, that lead to the increased use of physical restraints and explore avenues to reduce this use in future pandemics.

## Acknowledgments

We thank all the staff members and all the participating acute care hospitals.

## Author Contributions

**Conceptualization:** Takuya Okuno, Hisashi Itoshima, Yuichi Imanaka.

**Data curation:** Takuya Okuno, Jung-ho Shin, Susumu Kunisawa, Yuichi Imanaka.

**Formal analysis:** Takuya Okuno, Hisashi Itoshima, Jung-ho Shin.

**Funding acquisition:** Yuichi Imanaka.

**Methodology:** Takuya Okuno, Jung-ho Shin, Tetsuji Morishita.

**Project administration:** Takuya Okuno, Jung-ho Shin.

**Resources:** Takuya Okuno.

**Software:** Takuya Okuno, Tetsuji Morishita, Susumu Kunisawa.

**Validation:** Tetsuji Morishita.

**Visualization:** Takuya Okuno.

**Writing – original draft:** Takuya Okuno.

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
