## [Decision Letter · Decision Letter 0]

9 Aug 2021

PONE-D-21-19437

Physical restraint of dementia patients in acute care hospitals during the COVID-19 pandemic: A cohort analysis in Japan

PLOS ONE

Dear Dr. Imanaka,

Thank you for submitting your manuscript to PLOS ONE. After careful consideration, we feel that it has merit but does not fully meet PLOS ONE’s publication criteria as it currently stands. Therefore, we invite you to submit a revised version of the manuscript that addresses the points raised during the review process.

The manuscript is well assessed by two reviewers; however, several revisions are required in the present form. See the reviewers' comments carefully and respond them appropriately.

We look forward to receiving your revised manuscript.

Kind regards,

Masaki Mogi

Academic Editor

PLOS ONE

Journal Requirements:

2. Please report the codes used for extracting the outcome of interest (use of physical restraints) from the database

Reviewers' comments:

Reviewer's Responses to Questions

**Comments to the Author**

1. Is the manuscript technically sound, and do the data support the conclusions?

Reviewer #1: Partly

Reviewer #2: Yes

2. Has the statistical analysis been performed appropriately and rigorously? 

Reviewer #1: Yes

Reviewer #2: Yes

3. Have the authors made all data underlying the findings in their manuscript fully available?

Reviewer #1: Yes

Reviewer #2: Yes

4. Is the manuscript presented in an intelligible fashion and written in standard English?

Reviewer #1: Yes

Reviewer #2: Yes

5. Review Comments to the Author

Reviewer #1: In the background of the study, please describe how dose COVID-19 affect the care of patient with dementia in the hospital? is there visitation restriction? Isolation policy? Relocation of patient with dementia due to COVID case segregation? Manpower shortage? Dose hospital without COVID affected by those regulations?

Please explain what is "136 Hp with 97,233 admissions"

Please explain and elaborate why BMI is choosen as one of the parameter to group patients?

In figure 2, before COVID, hospital takes COVID cases already have higher percentage of restraint in the hospital. Is it because the hospital is less well trained in dementia care? No geriatric specialist? If a hospital already prone to restraint dementia patient (lack of geriatric culture), COVID -19 will only aggregate it. Will be good to explain as this may make the comparison between the hospitals become unfair.

Reviewer #2: General comments

The authors conducted a retrospective cohort study to assess the impact of regulatory changes, consequent to the pandemic, on physical restraint use among elderly dementia patients in acute care hospitals in Japan. Main finding suggests elderly dementia patients who require personal care might be more likely to be physically restrained during the COVID-19 pandemic in hospitals receiving COVID-19-positive patients. The result was notable and very interesting. However, I have some concerns that should be addressed regarding the methodology and interpretation of results.

Specific comments

Major

It would be better to provide more detailed definitions of the state of emergency declaration and Group 1/2. Is the state of emergency nationwide or only for a specific region? Why was the deadline set at July 4, 2020? Did the group 1 receive COVID-19 patients only after the state of emergency was declared or not regardless of the time of declaration?

L.195-205 This second paragraph is a general statement and does not seem to be relevant to the results of this study. Therefore, by changing this paragraph to a comparison of this study with past studies on physical restraint, the authors can make this study more unique. For example, comparing the study of physical restraint with other social changes such as endemics or epidemics of other infectious diseases, disasters, or terrorism.

L.209-222 This paragraph is the reason why physical restraint increased after the COVID-19 pandemic. It does not explain why group 1 has more physical restraints than group 2. This is because if the state of emergency had been declared at the national level, group 2 would have been recommended the same social distance as group 1. If it is due to social distance, the authors should divide the hospitals according to whether they are in an epidemic area or not, not whether they accept COVID-19 patients or not. Smaller hospitals may not be accepting COVID-19 patients even in epidemic areas. It would be better to reconsider the difference in patient factors between Group 1 and Group 2.

L. 223-241 The restriction of visits seems to be done regardless of the acceptance of COVID-19 patients. As stated in L237-240, the changes in the nursing side seems to be a strong factor. It may be better to move L237-240 to the first half of this paragraph and elaborate further.

Minor

It may be easier for readers to read the abstract if it is divided into Introduction, methods, results, and conclusion.

6. PLOS authors have the option to publish the peer review history of their article (what does this mean?). If published, this will include your full peer review and any attached files.

Reviewer #1: No

Reviewer #2: No

---

## [Author Response · Author response to Decision Letter 0]

2 Oct 2021

Journal Requirements:

PONE-D-21-19437

Physical restraint of dementia patients in acute care hospitals during the COVID-19 pandemic: A cohort analysis in Japan

PLOS ONE

Dear Dr. Imanaka,

Response: Thank you for this reminder. I have checked the manuscript and made sure that it complies with the style requirement of your journal.

2. Please report the codes used for extracting the outcome of interest (use of physical restraints) from the database 

Response:

The "code" is the Japanese medical fee code and is standardized for all medical institutions. Patients who are judged to have cognitive dysfunction and need special care are given an additional fee for dementia care; however, if physical restraint is used, the additional fee for dementia care is multiplied by 0.6.

Lines 121-122

The following has been added: “Data regarding the use of physical restraint during dementia care were extracted from the payment codes for services.”

＊Changes of reference list

Reference 9: remove the previous reference and replace it with relevant current reference

Reference 31: A relatively new report about physical restraining during the COVID-19 pandemic was added, which could be supportive of the result of this study.

Reviewer #1: 

Q.　In the background of the study, please describe how dose COVID-19 affect the care of patient with dementia in the hospital? is there visitation restriction? Isolation policy? Relocation of patient with dementia due to COVID case segregation? Manpower shortage? Dose hospital without COVID affected by those regulations?

Response: Lines 48-52

Thank you for your comment.

The following has been added: “Social distancing and limiting family visits impacted the hospital care systems in many ways, such as reduced communications with medical staff and family members; these in turn could exacerbate progressive cognitive dysfunction …”

A more detailed description has also been added in the Discussion section regarding this point.

Q. Please explain what is "136 Hp with 97,233 admissions"

Response：Thank you for this comment. I had quoted the number of hospitals and admissions included in our study, which was also shown in Figure 1. However, in the manuscript, I think mentioning the number of hospitals is not absolutely essential and may create confusion as pointed out. Therefore, I have deleted the number of hospitals (Hp).

Q. Please explain and elaborate why BMI is chosen as one of the parameter to group patients?

Response：Thank you for this comment. In order to be able to more easily understand the distribution and missing information from the Table, we categorized the BMI. However, BMI information, including missing or not, is unimportant in the context of this study; hence, it was removed from Table 1.

Q. In figure 2, before COVID, hospital takes COVID cases already have higher percentage of restraint in the hospital. Is it because the hospital is less well trained in dementia care? No geriatric specialist? If a hospital already prone to restraint dementia patient (lack of geriatric culture), COVID -19 will only aggregate it. Will be good to explain as this may make the comparison between the hospitals become unfair.

Response：Lines 255-265

The following has been added: “As shown in Figure 2, hospitals with COVID-19-positive cases already have higher percentages of restraint in the hospital than those before the COVID-19 pandemic. It may be desirable to evaluate the quality of dementia care and the involvement of a geriatric specialist; however, this is not possible owing to a limited database. However, the target population for this study was the inpatients for whom dementia care benefit was calculated. We believe that the quality of care in the two populations is secured to a certain extent because the hospitals need to have staff trained in dementia care and conduct regular care meetings in order to calculate the additional fee. Additionally, since this study uses the impact of COVID-19 as an intervention point to compare the percentage of physical restraint practices in the hospital for each group over time, we believe that this is not a problem.”

(Supplement): Owing to lack of data, I am unable to present the quality of care; nevertheless, I have information regarding the presence of the dementia care team. 

Group 1 January 2019 to February 2020 March 2020 to June 2020

Number of patients 80,468 16,765

Cared by dementia care team, n (%) 36193 (45.0) 9127 (54.4) 

Group 2 January 2019 to February 2020 March 2020 to June 2020

Number of patients 48,424 10,199

Cared by dementia care team, n (%) 13719 (28.3) 3369 (33.0) 

Reviewer #2: General comments

Specific comments

Major

Q. It would be better to provide more detailed definitions of the state of emergency declaration and Group 1/2. Is the state of emergency nationwide or only for a specific region? Why was the deadline set at July 4, 2020? Did the group 1 receive COVID-19 patients only after the state of emergency was declared or not regardless of the time of declaration?

Response：Thank you for this comment. Since this study targeted the first wave, which caused unprecedented disruptions, the study period was up to July, when the reemergence of the infection occurred (July 4 was chosen because we thought it would be preferable to aggregate the data on a weekly basis).

Lines 47-50

The following has been added: “To manage the pandemic, the Japanese government announced hospital restrictions, including those pertaining to family visits, at the end of March 2020. Eventually, a state of emergency was declared for specific areas on 7th Aprill 2020, and implemented nationwide on 16th April, 2020.”

Lines 128-133

The following has been added: “In Japan, hospitals that can accept COVID-19-positive patients were designated by the MHLW. If no COVID-19-positive patients were hospitalized during this study period, we considered that the impact of COVID-19 was small. Therefore, we categorized the study population into two groups: hospitals having at least one COVID-19-positive patient admission during the study period (Group 1) and those having none (Group 2).”

Q. L.195-205 This second paragraph is a general statement and does not seem to be relevant to the results of this study. Therefore, by changing this paragraph to a comparison of this study with past studies on physical restraint, the authors can make this study more unique. For example, comparing the study of physical restraint with other social changes such as endemics or epidemics of other infectious diseases, disasters, or terrorism. 

Response: Lines 204-214

The following has been added: “Dementia has increasingly gained importance as a public health concern, and the medical staff in acute care hospitals often needs to provide dementia care to elderly patients [25, 26]. Physical restraint, which is preferably avoided wherever possible, in conformance with worldwide recommendations, is often exercised in acute care settings, especially for elderly patients and those with dementia [7,8,15,27-29]. Physical restraint is exercised to prevent falls and self-extubation owing to the low availability of medical staff and inadequate resources to constantly monitor at-risk patients because of the immense workload [6,30]. There are few reports about changes in the implementation rate of physical restraint due to disasters such as the COVID-19 pandemic. However, one recent observational study showed the possibility of increased use of physical restraint during the COVID-19 pandemic [31], which supports our results.”

Q. L.209-222 This paragraph is the reason why physical restraint increased after the COVID-19 pandemic. It does not explain why group 1 has more physical restraints than group 2. This is because if the state of emergency had been declared at the national level, group 2 would have been recommended the same social distance as group 1. If it is due to social distance, the authors should divide the hospitals according to whether they are in an epidemic area or not, not whether they accept COVID-19 patients or not. Smaller hospitals may not be accepting COVID-19 patients even in epidemic areas. It would be better to reconsider the difference in patient factors between Group 1 and Group 2.

Q. L. 223-241 The restriction of visits seems to be done regardless of the acceptance of COVID-19 patients. As stated in L237-240, the changes in the nursing side seems to be a strong factor. It may be better to move L237-240 to the first half of this paragraph and elaborate further.

Response: The above two questions are related points from the same paragraph and, hence, I would like to respond to them together.

We believe that there could be two reasons for the increase in the use of physical restraint: 1) patient factors due to the spread of COVID-19, and 2) hospital (medical staff) factors. However, as pointed out by the reviewer, reason 1) is not in accordance with the results of this study. Therefore, as pointed out by the reviewer, we focused our discussion on the hospital (medical staff) factors only. As one of the reasons for considering the hospital (medical staff) factor to be strong, I would like to add that although it has been reported that cognitive function of the elderly may worsen with social distancing being implemented nationwide in Japan, we could not point out a reason for the obvious increase in the use of physical restraint in the ITS analysis at the hospitals without any hospitalization of COVID-19-positive patients in this study.

Lines 215-221

The following has been added: “We believe that the main reason for the significantly increased use of physical restraints for elderly dementia patients in only Group 1 during the COVID-19 pandemic was due to factors associated with the quality of care. Although it has been reported that cognitive function of the elderly may worsen with social distancing being implemented nationwide in Japan [32,33], the reason for an obvious increase in the use of physical restraint in the ITS analysis at the hospitals without any hospitalization of COVID-19-positive patients in this study was unclear.”

Lines 225-228

The following has been added: “Owing to the increase in nosocomial infections from February to April, 2020, medical staff were seen as epicenters, and this led to widespread irrational prejudice and discrimination against them in off duty-hours. They were denied use of public vehicles and their children were asked to refrain from attending nursery schools [36].”

Lines 235-242

The following has been added: “Factors associated with the care system, including limiting family visits, might have also possibly affected the result. In Japan, even the state of emergency is not legally binding; therefore, the hospital visit restrictions at hospitals without COVID-19 positive patients might have been more permissive than hospitals with COVID-19-positive patients’ hospitalizations. For dementia patients, communicating with visitors, especially family members, is important to maintain their cognitive function [29,31,38,39]. The Centers for Disease Control and Prevention guidelines allow care partners to visit patients if they are essential to the patients’ physical or emotional well-being, even during the COVID-19 pandemic [40].”

Minor

It may be easier for readers to read the abstract if it is divided into Introduction, methods, results, and conclusion.

Response

As per your suggestion, the abstract has been divided into sub-headings.

---

## [Decision Letter · Decision Letter 1]

10 Nov 2021

Physical restraint of dementia patients in acute care hospitals during the COVID-19 pandemic: A cohort analysis in Japan

PONE-D-21-19437R1

Dear Dr. Imanaka,

We’re pleased to inform you that your manuscript has been judged scientifically suitable for publication and will be formally accepted for publication once it meets all outstanding technical requirements.

Kind regards,

Masaki Mogi

Academic Editor

PLOS ONE

Additional Editor Comments (optional):

No further comment.

Reviewers' comments:

Reviewer's Responses to Questions

**Comments to the Author**

1. If the authors have adequately addressed your comments raised in a previous round of review and you feel that this manuscript is now acceptable for publication, you may indicate that here to bypass the “Comments to the Author” section, enter your conflict of interest statement in the “Confidential to Editor” section, and submit your "Accept" recommendation.

Reviewer #2: All comments have been addressed

2. Is the manuscript technically sound, and do the data support the conclusions?

Reviewer #2: Yes

3. Has the statistical analysis been performed appropriately and rigorously? 

Reviewer #2: Yes

4. Have the authors made all data underlying the findings in their manuscript fully available?

Reviewer #2: Yes

5. Is the manuscript presented in an intelligible fashion and written in standard English?

Reviewer #2: Yes

6. Review Comments to the Author

Reviewer #2: (No Response)

7. PLOS authors have the option to publish the peer review history of their article (what does this mean?). If published, this will include your full peer review and any attached files.

Reviewer #2: No

---

## [Editor Report · Acceptance letter]

12 Nov 2021

PONE-D-21-19437R1 

Physical restraint of dementia patients in acute care hospitals during the COVID-19 pandemic: A cohort analysis in Japan 

Dear Dr. Imanaka:

I'm pleased to inform you that your manuscript has been deemed suitable for publication in PLOS ONE. Congratulations! Your manuscript is now with our production department. 

Kind regards, 

on behalf of

Dr. Masaki Mogi 

Academic Editor

PLOS ONE